# Osteoarthritis Pain

**DOI:** 10.3390/ijms23094642

**Published:** 2022-04-22

**Authors:** Huan Yu, Tianwen Huang, William Weijia Lu, Liping Tong, Di Chen

**Affiliations:** 1Faculty of Pharmaceutical Science, Shenzhen Institute of Advanced Technology, Chinese Academy of Sciences, Shenzhen 518055, China; huan.yu@siat.ac.cn (H.Y.); wwlu@hku.hk (W.W.L.); 2Research Center for Computer-Aided Drug Discovery, Shenzhen Institute of Advanced Technology, Chinese Academy of Sciences, Shenzhen 518055, China; 3CAS Key Laboratory of Brain Connectome and Manipulation, Shenzhen-Hong Kong Institute of Brain Science-Shenzhen Fundamental Research Institutions, Shenzhen Institute of Advanced Technology, Chinese Academy of Sciences, Shenzhen 518055, China; tw.huang@siat.ac.cn; 4Department of Orthopaedics and Traumatology, The University of Hong Kong, Hong Kong SAR 999077, China

**Keywords:** osteoarthritis, pain, NGF/TrkA, CGRP, CCL2/CCR2, TNF-α, IL-1β, NLRP3, β-catenin

## Abstract

Joint pain is the hallmark symptom of osteoarthritis (OA) and the main reason for patients to seek medical assistance. OA pain greatly contributes to functional limitations of joints and reduced quality of life. Although several pain-relieving medications are available for OA treatment, the current intervention strategy for OA pain cannot provide satisfactory pain relief, and the chronic use of the drugs for pain management is often associated with significant side effects and toxicities. These observations suggest that the mechanisms of OA-related pain remain undefined. The current review mainly focuses on the characteristics and mechanisms of OA pain. We evaluate pathways associated with OA pain, such as nerve growth factor (NGF)/tropomyosin receptor kinase A (TrkA), calcitonin gene-related peptide (CGRP), C–C motif chemokine ligands 2 (CCL2)/chemokine receptor 2 (CCR2) and tumor necrosis factor alpha (TNF-α), interleukin-1beta (IL-1β), the NOD-like receptor (NLR) family, pyrin domain-containing protein 3 (NLRP3) inflammasome, and the Wnt/β-catenin signaling pathway. In addition, animal models currently used for OA pain studies and emerging preclinical studies are discussed. Understanding the multifactorial components contributing to OA pain could provide novel insights into the development of more specific and effective drugs for OA pain management.

## 1. Introduction

Osteoarthritis (OA) is one of the most common pain-evoking and disabling diseases globally affecting about 250 million people [1]. It severely affects the patients’ quality of life and imposes a great economic burden on individuals, families, and the entire society. With the aging and increasing life expectancy of the population, OA poses a major challenge to social and public health [2]. OA is a whole-joint disease involving all joint tissue types, namely, cartilage, menisci, synovial membrane, infrapatellar fat pads, and subchondral bone caused by the combined actions of systemic-susceptibility and local factors [3]. Varieties of factors, such as mechanical, inflammation, aging, and metabolic, are involved in the pathogenesis of OA, and ultimately lead to articular structural destruction and loss of functions of synovial joints, along with long-term chronic pain [4,5,6].

Pain is the main symptom of OA and the main reason for patients to seek medical help, and is defined by the International Association for the Study of Pain (IASP) as “an unpleasant sensory and emotional experience associated with, or resembling that associated with, actual or potential tissue damage” [7,8,9]. It is often regarded as an important warning signal that plays a protective role in the response to acute tissue injury and inflammation. When acute pain is not relieved and transitions into chronic pain, pain control becomes much more challenging [10,11]. OA patients often experience a dull, aching pain or unpredicted intensity of intermittent pain that is initially activity-related and subsequently more constant over time, representing one of the most prevalent and disabling chronic conditions [12,13]. In addition to the pain itself, long-term chronic pain has negative effects on mental health, sleep, and social participation [14,15]. Traditionally, NSAIDs, acetaminophen, and opioid analgesics are the most common prescription drugs for the management of OA pain. However, because the pathogenesis of OA pain is not well-understood, the current strategy for the intervention of OA pain cannot provide satisfactory pain relief, and the chronic use of the drugs for pain management is often associated with significant side effects and toxicities [16,17]. Unravelling the mechanisms of these multifactorial components contributing to the generation and maintenance of OA pain offers critical new insights into the development of more effective and safer pain treatment.

In this review, we mainly focus on the characteristics and mechanisms of OA pain, and important pathways associated with OA pain, such as nerve growth factor (NGF)/tropomyosin receptor kinase A (TrkA), calcitonin gene-related peptide (CGRP), C–C motif chemokine ligands 2 (CCL2)/chemokine receptor 2(CCR2) and tumor necrosis factor alpha (TNF-α), interleukin-1beta (IL-1β), the NOD-like receptor (NLR) family, pyrin domain–containing protein 3 (NLRP3) inflammasome, and the Wnt/β-catenin signaling pathway. Animal models currently used for OA pain studies and emerging preclinical studies are also discussed according to recent findings.

## 2. Materials and Methods

This narrative review search was conducted to describe the characteristics and mechanisms of OA pain, and of a variety of critical signaling pathways associated with OA pain, including NGF/TrkA, CGRP, CCL2/CCR2 and TNF-α, IL-1β, NLRP3 inflammasome, and Wnt/β-catenin. We also included common animal models used for OA pain studies and the results of emerging preclinical studies with limitations in terms of the English language. Original papers and reviews that evaluated the mechanisms of OA pain, OA pain-related animal models, and clinical efficacy for the treatment of OA pain were considered. In March 2022, two electronic databases, PubMed and Google Scholar, were searched with the terms osteoarthritis, pain, NGF/TrkA, CGRP, CCL2/CCR2, TNF-α, IL-1β, NLRP3, and Wnt/β-catenin. In particular, the following terms were applied to the search: (osteoarthritis AND pain), (osteoarthritis AND (NGF OR TrkA)), (osteoarthritis AND CGRP), (osteoarthritis AND CGRP), (osteoarthritis AND (CCL2 OR CCR2)), (osteoarthritis AND TNF-α), (osteoarthritis AND IL-1β), (osteoarthritis AND NLRP3), (osteoarthritis AND (Wnt OR β-catenin)), and (osteoarthritis AND animal model). Duplicates and nonrelevant records were excluded. Other original articles, reviews, and important publications were referred to in the references of the searched articles from database.

## 3. Characteristics and Structural Correlation of OA Pain

OA pain is the main symptom of the disease, but the true source of OA pain is unclear. Although the hallmark pathology of OA is articular cartilage lesion, no nerves and blood vessels exist in articular cartilage; thus, cartilage cannot directly generate OA pain [18]. However, the surrounding tissue types of a joint, including the synovium, fat pad, ligaments, joint capsule, and subchondral bone, are all highly innervated by rich sensory and sympathetic nerves, and can be the source of nociceptive pain. Previous studies showed that OA pain is linked with synovitis, and alterations in pain are related to changes in synovitis [19]. Osteoclasts also induce sensory nerve innervation in subchondral bone, which is the cause of OA pain [20,21]. In addition, abnormal infrapatellar fat pads, defects in cartilage and subchondral bone microfractures, synovitis, osteophyte formation, and intraosseous hypertension were all associated with OA pain [13,22,23,24,25,26]. Nociceptor nerve terminals are key molecular transducers of these noxious stimuli that express varieties of ion channels such as members of the transient receptor potential (TRP) ion channel family including TRP vanilloid 1 (TRPV1), TRP melastatin 3 (TRPM3), TRP ankyrin 1 (TRPA1), and voltage-gated sodium channel members, including Nav1.8 and Nav1.7 [27,28] (Figure 1A). When detecting potential noxious stimuli such as mechanosensations and chemical or thermal sensations, these terminals are activated to evoke pain transmission and release neuropeptides including CGRP and substance P (SP) [29]. Subsequently, pain signals are transmitted along the dorsal root ganglia (DRG), where the cell body of sensory neurons and the dorsal horn of the spinal cord reside, leading to the upregulation of CCL2, NLRP3, and Wnt/β-catenin (Figure 1B,C). Second-order neurons within the dorsal horn of the spinal cord are activated through neurotransmitters including CGRP, SP, and glutamate [28,30] (Figure 1C). Ultimately, ascending pathways activate the higher central nervous system, leading to the conscious awareness of pain [31,32].

OA pain was long thought to be nociceptive pain triggered by damaged or inflamed tissue. However, judging from the current evidence, OA pain is likely driven by both nociceptive and neuropathic mechanisms [32]. First, the distribution of sensory nerve fibers in various parts of the joint is the material basis for the body to perceive pain. OA pain is usually located at the affected joint, which is related to exercise and weight bearing, and relieved at rest, suggesting that OA pain can be caused by sensing structural damage [33]. Second, although OA is not typical inflammatory arthritis, inflammation mediators play a key role in OA pathogenesis [34,35]. Various proinflammatory factors are produced in OA pathological states and released into the joint fluid, driving a series of cascading reactions. Cytokines such as IL-1β, IL-6, TNF-α, and NGF, and chemokines including CCL2 and fractalkine can participate in the generation of pain and peripheral sensitization by interaction with sensory nerves [36]. The presence of persistent pain inputs in patients with OA leads to the sensitization of the central and peripheral nervous systems, leading to mechanical tenderness and hyperalgesia that are typical of neuropathic pain around the OA joints, supporting the idea that neuropathic pain components exist in OA pain [30,31]. Central sensitization is likely involved in patients experiencing pain who suffered severe OA pain despite the less severe joint pathology [37]. Other evidence showed that increased synovial NGF expression, and higher cerebrospinal fluid levels of serotonin and dopamine metabolites are positively correlated with pain severity and central sensitization in OA patients [38]. Increased central sensitization is also associated with pain at rest, and affected postoperative pain persistence and dissatisfaction following arthroplasty [39,40]. However, the mechanisms of central sensitization underlying OA pain remain undefined.

## 4. Mechanisms of OA Pain

### 4.1. NGF/TrkA Signaling

NGF was first discovered approximately 60 years ago as an indispensable neurotrophic factor that induces sensory and sympathetic nerve growth [41]. It is now known that NGF is also a proinflammatory cytokine mediating pain signal transduction that leads to the sensitization of the nervous system. NGF is upregulated by cytokines such as TNF and can be produced by non-neural cells such as macrophages, mast cells, synoviocytes, and neutrophils, found in the joint tissue of individuals with OA [42,43]. NGF acts on its high-affinity receptor, TrkA, which is then taken up by endosomes and undergoes long-distance retrograde transport from the distal axon to the cell body of DRG. The activation of NGF/TrkA signaling results in the upregulation and release of inflammatory mediators, and the increased activation of TRPV1 and Nav1.8, leading to a broad increase in neuronal activity. Moreover, NGF upregulates the expression of neurotransmitters such as CGRP, substance P, and brain-derived neurotrophic factor, which contribute to central pain sensitivity [44].

McNamee and colleagues found that the induction of NGF expression following the destabilization of medial meniscus (DMM)-induced OA was correlated with pain-related behavioral changes during OA development in mice [45]. Ohashi et al. identified that a subtype of macrophages, CD14 (high), in the synovium of hip OA patients expressed higher levels of NGF, which contribute to joint pain and central sensitization [46]. Our previous study demonstrated that DMM surgery increased NGF/TrkA expression and retrograded transport of NGF in DRG in mice. Neurite outgrowth and NGF/TrkA signaling are critical drivers for OA hyperalgesia in both experimental OA animal models and in OA patients experiencing pain, which was unrelated with the severity of cartilage degeneration [47]. More recently, we demonstrated reduced OA pain, but worsened articular cartilage lesions and osteophyte outgrowth in DMM-induced mice OA with CRISPR-mediated ablation of NGF [48]. The results of rapidly progressive cartilage destruction were also observed when anti-NGF antibodies were used to treat OA patients in a clinical trial. Thus, it is intriguing how NGF function differentially induces joints pain and pathology. A more thorough investigation for the role of NGF in OA would provide new insights into the development of new medications with high efficacy for the treatment of OA pain while reducing toxicity.

### 4.2. CGRP

CGRP is a 37-amino acid neuropeptide that is widely expressed in the peripheral and central nervous systems, with the most abundant in nociceptive neurons, e.g., in the DRG and nerve fibers that project to the dorsal horn of the spinal cord [49,50]. As a sensory neuropeptide, CGRP can dilate blood vessels, affect peripheral pain sensitization and inflammation, and play a critical role in the process of neurogenic inflammation and pain generation [51]. Many studies showed that CGRP specifically targets the pathologic pain response associated with sensitization [52,53]. CGRP plays an important role in the physiology of migraine, and CGRP receptor antagonists could block migraines and is the most successful translational product for migraine treatment [54]. We found a critical role of CGRP during bone fracture healing by mediating the interaction between nervous and musculoskeletal systems [55]. Stockl and their colleagues analyzed the influence of exogenous αCGRP and SP in vitro on the chondrocyte metabolism and modulation of associated signaling pathways from OA patients receiving total knee arthroplasty and non-OA cartilage donors. The study found that both αCGRP and SP treatment promoted apoptosis and senescence, and decreased the expression of chondrogenic markers in OA chondrocytes via the activation of ERK signaling, but αCGRP enhanced chondrocyte anabolism and had protective effect on healthy chondrocytes; SP had minimal effects on chondrocytes from healthy cartilage [29]. However, the precise role of CGRP in OA pain remains unexplored.

Clinical evidence shows that serum CGRP levels and CGRP-positive nerve fiber density are correlated with pain symptoms and disease severity in OA patients [56]. CGRP receptor antagonists and CGRP neutralizing antibodies could alleviate pain in OA animal models induced by different methods [57]. Recent studies also showed that blocking the activity of CGRP through the administration of receptor antagonists or the deletion of CGRP could prevent sensitization and mechanical allodynia caused by inflammation and neuropathic pain [58,59]. These results indicate that CGRP is an effective target for OA management. However, a recent Phase II clinical trial showed that CGRP antibodies lack efficacy on moderate and severe OA pain (NCT02192190) [60]. The reason for these negative results needs further investigation. A study analyzed CGRP and its receptor expression in synovial tissue harvested from male and female patients receiving total knee arthroplasty [61]. Synovial CGRP expression was positively correlated with pain severity in women but not in men, whereas the expression of RAMP1 was not correlated with pain scores in either men or women. These differential CGRP and RAMP1 expression levels by sex suggest that different pain mechanisms exist in men and women with knee OA.

### 4.3. CCL2/CCR2

CCL2, also called monocyte chemo attractant protein (MCP)-1, was first identified in 1989 and is described as a “tumor-derived chemotactic factor” [62]. It can be produced by many activated cells, including macrophages, endothelial cells, fibroblasts, and lymphocytes, and functions as a potent chemoattractant for a variety of immune cells such as monocytes, macrophages, dendritic cells, and memory T cells in multiple inflammatory diseases [63,64]. The development and maintenance of OA pain behavior such as mechanical allodynia and movement-evoked pain involve molecular changes in the sensory neurons of the DRG, including alterations of chemokines and their receptors in mRNA and protein expression. CCL2/CCR2 signaling was recently found to be central to the development of knee OA pain [65]. In that study, CCL2/CCR2 mRNA and protein expressions in L3–L5 DRG neurons were elevated at 8 weeks postsurgery, and returned to normal levels at 16 weeks after DMM surgery. Mechanical allodynia in mice that lacked CCR2 was resolved from 8 weeks onwards, while at 8 weeks, *Ccr2*-null mice did not display movement-provoked pain behaviors, and showed fewer macrophage infiltrations in DRG. These results demonstrated that neuronal CCL2 and CCR2 from DRG play vital roles in mediating macrophage infiltration and OA pain sensitivity. A study found that mice lacking CCL2 or CCR2 were protected against OA with reduced monocyte or macrophage numbers in joint tissue [66]. Blocking CCL2/CCR2 signaling also markedly attenuated OA-like phenotypes, such as macrophage accumulation, synovitis, and cartilage lesions in mouse OA. These data demonstrated that monocytes recruited via CCL2/CCR2 promoted inflammation and cartilage damage in OA. Another experimental OA study found that local CCL2/CCR2 signaling in the joint caused knee hyperalgesia through the direct stimulation of intra-articular CCR2 positive sensory nerves [67]. In the study, the intra-articular injection of recombinant CCL2 directly excited sensory afferents and caused knee hyperalgesia in wild-type mice, while the administration of a CCL2 receptor antagonist alleviated established hyperalgesia. Moreover, sensory neurons in the L4-DRG were excited by in vivo calcium imaging observed in DRG; the coexpression of CCR2 and sensory nerve markers such as PGP9.5 and CGRP was also found. These results suggest that CCL2/CCR2 signaling contributes to OA pain not only through recruiting monocytes or macrophages to the local joint and DRG, but also through the direct activation of sensory afferents. Targeting CCL2/CCR2 signaling may be a promising therapeutic approach for OA treatment.

### 4.4. TNF-α and IL-1β

Inflammation is involved in the initiation and development of OA [36,68]. Chronic and low-grade inflammation is distinct from that in prototypical inflammatory arthritis–rheumatoid arthritis. It affects the whole joint, resulting in joint pain, synovial hypertrophy, synovitis, the inflammation of the infrapatellar fat pad, accelerated cartilage loss, and osteophyte formation [13,69]. Varieties of proinflammatory cytokines, such as TNF-α, IL-1β and IL-6, are produced and primarily mediated by innate immune cells and, to a lesser degree, adaptive immune cells [70,71].

TNF-α and IL-1β are the most extensively studied proinflammatory cytokines involved in the pathophysiology of OA. Despite encouraging results from animal studies, anticytokine therapies in clinical trials have not achieved satisfactory pain relief in OA patients, revealing unclear mechanisms by which TNF-α and IL-1β mediate OA pain [72,73]. A study reported that concentrations of IL-1β in synovial fluid were inversely associated with knee pain; TNF-α was correlated with total WOMAC pain, and especially with pain during movement and at rest [68]. Li et al. found that the scores of the numeric rating scale were negatively correlated with levels of TNF-α, and the scores of visual analog scales were negatively correlated with the expression levels of IL-1β, IL-6, and TNF-α in the synovial fluid. They highlighted the importance of anti-inflammation therapy in the early stage of OA, when the expression of IL-1β, IL-6, and TNF-α is high [74]. TNF-α might be mainly responsible for both inflammatory pain and bone pathology. In addition to driving the chronic inflammation of the joint via NF-κB, TNF-α plays important roles in activating macrophages, and stimulating osteoclast proliferation and differentiation, which are closely related to OA pain. A previous study also showed that, in a monosodium iodoacetate (MIA)-induced mouse model of OA pain, both TNF-α and IL-6 expression increased in synovial tissue and joint capsules between Days 1 and 28, with the peak at Day 4 [75]. A single injection of TNF in the knee induced the persistent dose-dependent sensitization of peripheral sensory neurons, which was detected by mechanical allodynia; this effect was reversed by the simultaneous administration of etanercept or a COX inhibitor [76,77]. TNF-α is thus a promising therapeutic target with unknown underlying mechanisms for OA pain.

IL-1β is the other important proinflammatory cytokine involved in the chronic and low-grade inflammation in OA pathophysiology [35]. Many studies demonstrated that IL-1β is a critical cytokine upregulating catabolic and inflammatory pathways during cartilage and bone homeostasis by increasing the production of matrix-degradative enzymes in chondrocytes and promoting osteoclast differentiation [78]. Moreover, IL-1β increased expression of NGF which mediated pain genesis [79,80]. Inhibiting IL-1β and its interaction with cell surface receptors (IL-1LR) was proposed and investigated for the treatment of OA.

### 4.5. NLRP3 Inflammasome

The NLRP3 inflammasome is a member of innate immune system receptors and an important molecule involved in the regulation of active IL-1β. Its activation requires two phases [81]. In the priming phase, the expression of inactive NLRP3 and pro-IL-1β mRNA is increased via nuclear factor kappa B (NF-κB)-mediated transcriptional regulation. In the second phase, the assembly of the inflammasome is triggered resulting in the activation of caspase 1 and release of IL-1β and IL-18 [82]. Our previous study showed that the activation of inflammasomes results in the cleavage of caspase-1, which subsequently processes pro-IL-1β, and promotes the maturation and release of IL-1β, ultimately producing many proinflammatory cytokines such as IL-1β and TNF-α, and degradative enzymes, including MMP13 and Adamts5, which drive synovial inflammation and cartilage degradation [83]. Growing evidence demonstrates that the NLRP3 inflammasome is dysregulated during OA development and contributes to the generation of chronic pain. In a rat model of painful neuropathy, NLRP3 expression increased in the DRG, and the intrathecal injection of NLRP3 siRNA markedly prevented mechanical allodynia. In naive rats, the intrathecal injection of AAV-expressing NLRP3 could markedly decrease the paw withdrawal threshold [84]. Another study found the spinal microglia, and NLRP3 formation and activation contribute to opioid-prolonged neuropathic pain [85]. Cheng et al. found that dexmedetomidine improved pain symptoms and the cartilage lesions of papain-induced OA in rats through the inhibition of the NF-κB pathway and NLRP3 inflammasome [86]. Therefore, targeting inflammasome activity may be a novel and effective therapeutic strategy for OA pain.

### 4.6. Wnt/β-Catenin

Wnt/β-catenin signaling pathway is required for embryonic joint formation and adult bone homeostasis balance. Studies showed increased β-catenin expression in OA cartilage, and the overexpression of β-catenin in chondrocytes induces upregulated expression of matrix-degrading enzymes [87]. In addition, the hyperactivation of the Wnt/β-catenin signaling pathway induces a rapid increase in the production of matrix metalloproteinases and Adamts, leading to proteoglycan degradation [88]. We previously established β-catenin(Ex3)^Col2CreER^ mice in which β-catenin degradation was selectively inhibited, resulting in β-catenin accumulation in articular chondrocytes, and these conditional-activation mice eventually led to OA-like phenotypes, such as the progressive destruction of articular cartilage and osteophyte formation [89]. These studies demonstrated that Wnt/β-catenin plays a vital role in the pathogenesis of OA, and inhibiting β-catenin may be a novel approach to modify OA pathology. 

Wnt proteins are mainly expressed in mature neurons, and Wnt signaling plays a vital role in the pathophysiology of the nervous system, such as neurogenesis, neuroinflammation, and neurodegeneration [90,91,92,93]. The dysregulation of Wnt signaling is implicated in central nervous system injury, neurodegenerative diseases, and chronic pain. Accumulating evidence showed Wnt signaling disorder in DRG, the dorsal horn of the spinal cord, and sciatic nerves in rodent models mimicking chronic pain, which were reviewed by Zhou et al. [94]. Wnt3a and Wnt5a, two Wnt ligands that activate the canonical and atypical pathways, respectively, were markedly upregulated in the DRG and the dorsal horn of the spinal cord in different models of chronic pain, including inflammatory and neuropathic pain [95,96]. The administration of small-molecule drugs of Wnt agonists and ligands such as Wnt1 and Wnt5a induced mechanical allodynia in wild-type mice, which is abolished by Wnt inhibitors [96,97]. Studies demonstrated that blocking Wnt signaling pathways using NSC668036 (disheveled inhibitor), iCRT14, and XAV-939 (β-catenin inhibitor) reverses pain and pain-associated behaviors in different models of chronic pain [98,99]. Moreover, multiple studies suggest that the aberrant activation of Wnt signaling causes chronic pain through the increased expression of proinflammatory cytokines and chemokines, and enhances the activation of macrophages in the DRG, microglia, and astrocyte within the spinal-cord or brain regions, ultimately resulting in neuroinflammation and sensitization [95,100,101,102,103,104,105,106]. A study by Zhang et al. demonstrated that XAV-939 increased mechanical withdrawal thresholds through the downregulation of serum TNF-α and IL-18 in a rat DRG compression model [101]. Other studies found that XAV-939 decreased IL-1β in DRG in a rat neuropathic pain model [98]. Wnt inhibitors reduced CCL2 expression in the DRG of a rat neuropathic pain model, which plays a crucial role in macrophage migration and infiltration [98]. These findings indicate a neuroimmune regulatory role of Wnt signaling in the generation of chronic pain. Taken together, this evidence demonstrates that the aberrant upregulation of Wnt signaling contributes to chronic pain, and Wnt signaling could be a novel and promising target for chronic pain, including OA pain.

The Wnt/β-catenin pathway plays a critical role in the pathogenesis of both joint structure and chronic pain. Thus, targeting this pathway may represent a safe therapeutic intervention in relieving OA pain, reducing bone, cartilage, and synovial pathology. Regarding the limited understanding of Wnt signaling in the molecular mechanisms of OA pain, further studies may focus on how Wnt/β-catenin signaling regulates inflammatory cytokines and chemokines, and the crosstalk of Wnt/β-catenin signaling with the immune system.

## 5. Common Animal Models for OA Pain Research

Preclinical experimental OA models are important tools for exploring the pathogenesis of OA pain, and to evaluate the effects of targeted therapeutic interventions. Although several models, such as surgically, chemically, and genetically induced, spontaneous, and mechanical loading-induced OA models were developed for OA studies, only a few can definitely generate pain behaviors, and be utilized for studying the mechanism of OA pain [33,52,107]. Among them, the chemical MIA-induced OA and surgical models are the most popular. Miller et al. reviewed published papers about OA pain animal models and found that the MIA and surgical models accounted for 54% and 33% since 2008, respectively [108]. The MIA model was first recorded to induced an OA-like phenotype in 1987, and OA-related pain behaviors were known until 2003. This model induces a rapid, reproducible, robust painlike phenotype, and extensive joint pathology, which raises questions over its relevance to human OA [109]. The intra-articular injection of MIA leads to chondrocyte death through the inhibition of glyceraldehyde-3-phosphatase and disrupting cellular glycolysis, resulting in cartilage degeneration and bone destruction, which mimics certain aspects of OA pathology [110,111]. Comparison of the most commonly used animal models for OA pain were seen in Table 1.

The surgical destabilization of animal knees is performed to mimic mechanical-instability-induced human OA. This slowly progressive model is generally considered to be more ideal than the MIA model for exploring OA pathogenesis in early stages and during pharmacological intervention. The most commonly used surgical techniques are the transection of the anterior cruciate ligament (ACLT) and medial meniscal destabilization (DMM) [112]. The anterior cruciate ligament originates from the posterolateral wall of the femoral condyle, and enters the center of tibial plateau, which restricts the tibia from moving forward [113]. The ACLT-induced OA is a little more aggressive and severe than DMM. A previous study from Glasson et al. reported that ACLT is not recommended as a mouse model of OA, and that this procedure could develop severe OA and subchondral bone destruction [112,114]. DMM surgery is performed to induce OA by transecting the medial meniscotibial knee ligament causing joint destabilization in mice. This procedure is relatively easier to operate than ACLT and leads to OA-like phenotypes, such as the slowly progressive loss of articular cartilage, osteophyte formation, and synovitis, which is similar to those seen in sport injuries and aging-related OA [107,116,118]. Therefore, the DMM model is an important tool to investigate OA-associated mechanisms, pain, and therapies. There is evidence showing that joint destabilization results in an acute, transient pain phase associated with postoperative trauma and a later chronic OA pain phase [119]. In our lab, we characterized the pain pattern in a DMM-induced OA model, and found increased pain sensitivity and decreased spontaneous activity in mice after DMM surgery [117]. Recently, a study compared sensory innervation in the three models of rats above and found that the DMM and MIA models showed typical changes in mechanical hyperalgesia and cold hyperalgesia at Day 14 [115]. They also found that increased Netrin1, NGF, CGRP, and TRPV1 expression was observed in the DMM synovium at Day 14, and in the ACLT synovium at Day 28 compared with in MIA. CGRP and NGF expression in DRG was the highest in the DMM model. This study indicated that surgical modeling may be more useful for KOA pain research.

## 6. Emerging Preclinical Drugs Targeting OA Pain

Emerging pharmacological therapies in clinical trials with primary outcome measures of OA pain are summarized in Table 2.

### 6.1. Nerve Growth Factor Inhibitor

NGF/TrkA signaling once received great attention as a promising target for treating OA chronic pain since good results of pain reduction were reported in subjects with moderate-to-severe knee OA [120]. However, due to adverse events including rapidly progressive joint damage in patients receiving a treatment of anti-NGF antibodies, clinical trials of anti-NGF antibodies were suspended by the Food and Drug Administration (FDA) in US [135]. Phase 3 clinical trials of anti-NGF antibodies were resumed from 2015. Results of randomized and placebo-controlled Phase 3 trials of tanezumab and fasinumab were published in 2019, and further confirmed the efficacy of these two anti-NGF antibodies in relieving pain and improving joint function over 16 weeks in treating severe hip or knee OA (NCT02709486, NCT02528188, NCT02697773, NCT02447276) [121,122,123,124]. In Phase 3 trials of intravenous tanezumab versus oral NSAIDs for the treatment of knee or hip OA, tanezumab reduced pain, and improved function and global scores, especially in patients of poor analgesia with nonsteroidal anti-inflammatory drugs (NSAIDs) (NCT00863304, NCT00830063, NCT00744471) [125,126]. However, both drugs increased the risk of rapidly progressive joint damage leading to total joint replacement compared to the placebo group. NGF inhibitors could effectively relieve pain symptoms in OA patients, but their safety should be carefully evaluated. How exactly blocking NGF leads to rapid OA progression needs to be carefully investigated. Despite rapidly progressive joint damage occurring in knee OA trials, treatment with an NGF inhibitor showed positive results in lower back pain. The results of a Phase 3 trial demonstrated that fasinumab significantly improved lower back pain with progressive OA, observed in only few accompanied peripheral OA cases (NCT02620020) [127]. Blocking TrkA is the other strategy to inhibit NGF/TrkA signaling, probably with fewer adverse effects. The oral administration of ASP7962, a TrkA inhibitor did not reduce OA pain compared with naproxen in a Phase 2a clinical trial (NCT02611466) [128], However, the intra-articular injection of TrkA inhibitor GZ389988A greatly improved WOMAC pain and overall knee pain (NCT02845271) [136].

### 6.2. TNF Antibody

Erosive hand OA is the most aggressive subtype of OA, predominantly affecting women, and characterized by the articular inflammation and radiologically central erosion of the joint [137]. In patients with erosive hand OA, 40 mg subcutaneous injections of adalimumab showed no effect on pain, synovitis, or bone-marrow lesions with MRI-detected synovitis in comparison with a placebo after 12 weeks (ACTRN12612000791831) [129]. The result of a multicenter, randomized controlled trial (RCT) demonstrated that adalimumab administration was not superior to NSAIDs in decreasing pain scores in patients with hand OA (NCT00597623) [130]. Despite negative results in this nonweight-bearing small joint, positive effects of adalimumab were found in knee OA. In 2012, Maksymowych et al. reported that adalimumab significantly improved mean WOMAC pain and target joint swelling at 12 weeks in patients with knee OA (NCT00686439) [73]. An open-label RCT study evaluated adalimumab versus hyaluronic acid (HA) in patients with moderate-to-severe knee OA. Enrolled patients received an intra-articular injection of 10 mg adalimumab or 25 mg HA, followed by oral 200 mg/day celecoxib for 4 weeks. Pain decrease in VAS and WOMAC scores from Week 1 to 4 was markedly greater in the adalimumab than that in the HA group [138]. A Phase 2 double-blind RCT of subcutaneous injections of adalimumab compared to placebo in knee OA with clinical features of inflammation and persistent pain has just been completed (NCT02471118). The results should come out soon.

MEDI7352 is a monoclonal antibody specifically binding to NGF and TNF-α, thus blocking their effects. A Phase 1 clinical trial of MEDI7352 in patients with painful knee OA after subcutaneous or intravenous injection of single or multiple ascending doses was completed last year (NCT02508155). A Phase 2b RCT to evaluate the efficacy and safety of MEDI7362 in subjects with painful knee OA is ongoing (NCT04675034).

### 6.3. IL-1β Antibody

Two teams reported that IL1 receptor antagonist anakinra decreased VAS pain and global handicap in patients with severe erosive hand OA who had failed conventional treatment [139,140]. However, in a Phase 2a RCT, treatment with lutikizumab, a dual variable domain immunoglobulin neutralizing both IL-1α and IL-1β, failed to improve joint pain and function, and structural destruction in patients with erosive hand OA (NCT02384538) [131]. Chevalier et al. found that a single intra-articular injection of 50 mg or 150 mg anakinra in OA knee (NCT00110916) could not improve OA symptoms [72]. Results of a clinical trial using systemically administered AMG 108, an anti-IL-1R1 human IgG2 monoclonal antibody, demonstrated statistically insignificant improvement in the WOMAC pain scores of knee OA patients (NCT00110942) [132]. New strategies through gene therapy for IL-1Ra were developed for OA treatment. This approach leads to the sustained release of cytokines in local joints. The safety of a single intra-articular injection of recombinant adenoassociated virus carrying IL-1Ra was evaluated in patients with moderate knee OA in a Phase I clinical trial (NCT02790723). In addition to positive results in hand OA, treatment with lukitizumab (ABT-981) seemed to achieve good results in knee OA. Two Phase 1 studies demonstrated that treatment with lukitizumab was well-tolerated and reduced inflammation biomarkers in patients with mild-to-moderate knee OA (NCT 01668511) [141,142]. In a Phase 2 trial, lukitizumab markedly reduced WOMAC pain at Week 16 in patients with knee OA (NCT02087904) [133]. Recently, the Canakinumab Anti-inflammatory Thrombosis Outcomes Study (CANTOS) found that IL-1*β* inhibition with canakinumab had lower incidence rates of total joint replacement compared with the placebo group (NCT01327846) [143].

### 6.4. Wnt Inhibitor

A Phase 1 clinical trial reported that the intra-articular injection of novel Wnt signaling pathway inhibitor lorecivivint (SM04690) significantly alleviated the pain and improved the knee function of patients with OA (NCT04385303) [134]. Although the Phase 2a trial failed to meet the given primary endpoint, the administration of SM04690 was tested to be safe and well-tolerated, and the optimal effective dose of SM04690 was then identified by its Phase 2b trial [144,145]. Except for suppressing Wnt signaling, lorecivivint exhibited anti-inflammatory and chondroprotective activities in OA preclinical studies that seemed to be independent of β-catenin. Instead, they may have been mediated by the blockage of two important kinases, CLK2 and DYRK1A.

### 6.5. NLRP3 Inhibitor

IFM-2427 (DFV890) completed Phase 1 clinical trials in COVID-19 patients with pneumonia. Phase 2 clinical trials in COVID-19 patients with pneumonia are complete, but no results have been released yet. A Phase 2a proof-of-concept clinical trial is proceeding to determine the efficacy of oral administration of DFV890 vs. placebo in OA patients for relieving knee pain (NCT04886258). Dapansutrile™ (OLT1177™) is a new generation of oral NLRP3 inhibitor exhibiting great safety and good efficacy in acute gout treatment [146]. It has no effect on the priming phase of the NLRP3 inflammasome formation and on TNF-α expression. Despite no results having been achieved from topically applied OLT1177 gel in treating moderate-to-severe OA pain, OLT1177™ deserves further investigation on OA pain (NCT02104050).

### 6.6. Ion Channels Modulator

TRPV1 is a ligand-gated, voltage-gated, and nonselective cation channel that is abundantly expressed by sensory neurons. On the basis of their excellent pain control effects in vitro, TRPV1 modulators are attractive drugs that are undergoing clinical investigation for the treatment of OA pain. In 2018, CNTX-4975, a novel TRPV1 agonist, was granted fast-track designation for OA pain treatment by the US FDA. In a Phase 2 multicenter double-blind study, CNTX-4975 treatment achieved dose-dependent improvement in knee OA pain until 24 weeks after patients had received a single intra-articular injection of 0.5 mg CNTX-4975 or 1.0 mg CNTX-4975 or placebo [147]. To further evaluate the efficacy and safety of TRPV1 agonists and antagonists for OA pain treatment, larger trials are required, and the longer-term effects of these drugs need to be evaluated.

TRPA1 is a membrane-associated cationic channel that is widely expressed in neurons, and it mediates neurogenic and inflammatory pain to act as a sensor for toxic stimuli induced by exogenous compounds [148]. The abnormal activation of TRPA1 is closely associated with neuropathic pain, and TRPA1 inhibitors or gene deletions reduce painful behavior in OA mice [149]. A Phase 2 study to evaluate the safety or efficacy of LY3526318, a TRPA1 antagonist, in patients with knee OA pain is currently ongoing (NCT05080660).

## 7. Conclusions

OA pain involves complex peripheral and central mechanisms. Nerve sensitizations are major characteristics for pain transmission in OA patients that may contribute to the discordance between pain and joint pathology. NGF inhibitors reduce OA pain, but their safety should be carefully evaluated, and mechanisms underlying rapid OA progression should be further explored. Bidirectional interactions between the immune and nervous systems are recognized to be a major pathological mechanism of chronic pain. Neural CCL2/CCR2 signaling contributes to OA pain through recruiting both monocytes and macrophages to local joints, DRG, and the spinal cord, which represent new molecular neuroimmune regulatory mechanisms for the generation and maintenance of OA chronic pain. Despite the disappointing results from human trials, CGRP should be further investigated with subgroups of females and a higher expression of CGRP in serum or joint fluid. TNF-α is greatly involved in inflammatory and immune pathways, and bone pathology. TNF receptors include TNF receptor 1 (TNFR1) and TNFR2. These two receptors have distinct tissue localization and functions. TNFR1 is expressed ubiquitously in most cell types and functions primarily in mediating proinflammatory responses; TNFR2 is mainly expressed in immune, neuronal, and endothelial cells, and functions as an anti-inflammation and immunoregulatory factor. New drugs specifically targeting TNF receptors may be developed with safer and more effective properties. These urge us to better understand the exact role of TNF receptors in the pathogenesis of OA and pain. The NLRP3 inflammasome pathway plays an important role in the generation of OA pain. The activation of the NLRP3 inflammasome in chronic pain may occur more frequently in DRG and the spinal cord. Current knowledge about the role of Wnt signaling in OA is limited, and further studies may focus on how Wnt/β-catenin signaling regulates inflammatory cytokines and chemokines in the DRG and dorsal horn of the spinal cord, and the brain, and the crosstalk of Wnt/β-catenin signaling with immune system. Unravelling how these multifactorial components and their interactions function in the generation and maintenance of OA pain could provide novel insights into developing more specific and effective drugs for OA pain management.

## Figures and Tables

**Figure 1 ijms-23-04642-f001:**
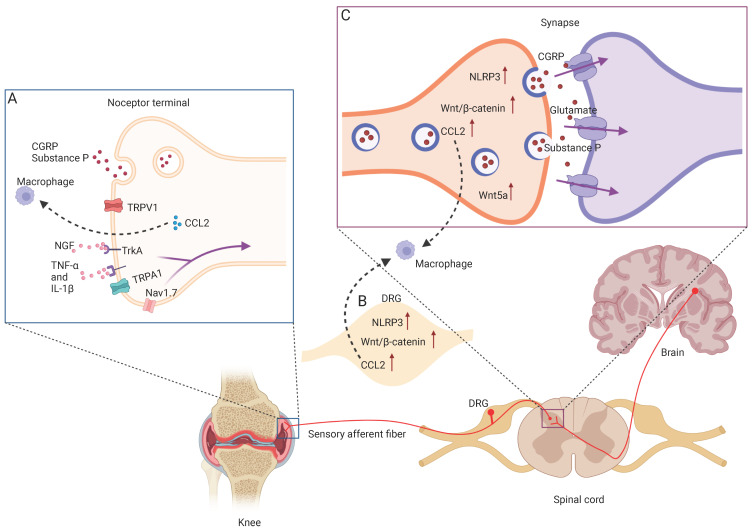
Overview of OA pain transmission and associated signaling pathways. (**A**). Peripheral terminals of nociceptors contain a variety of transducing channels that convert harmful stimuli into electrical activity, and thus action potentials in nociceptors that travel back to the central nervous system. In the event of painful stimulatory factors such as NGF, TNFα, and IL-1β acting on their receptors, ion channels such as TRPV1 and Nav1.7 are activated to transmit a pain signal and CCL2 expression, and are upregulated to recruit macrophage. (**B**). Cell bodies of nociceptors are located in the dorsal root ganglion. Increased expression of NLRP3, Wnt/β-catenin, CCL2, Wnt5a in DRG in chronic pain states. (**C**). Nociceptive signals are transmitted at a central synapse in the spinal cord through the release of a variety of excitatory neurotransmitters, such as glutamate, CGRP, or substance P, which could excite second-order nociceptive projection neurons. NLRP3, Wnt/β-catenin, CCL2, and Wnt5a expression is upregulated in presynaptic neurons.

**Table 1 ijms-23-04642-t001:** Comparison of the most commonly used animal models for OA pain.

Model	Species	Procedure	Mechanism of Model	Disease Onset	Advantages	Disadvantages	References
MIA	Rat, mouse	Intra-articular injection of MIA	Disrupt chondrocyte glycolysis via inhibiting glyceraldehyde-3-phosphatase dehydrogenase	1 week	Rapid, reproducible, robust pain-like behaviour and peripheral/central sensitization partially characterized.	Extensive and rapidly developing pathology does not mimic human OA.	[109,110,111,112]
ACLT	Dog, Rat	Transection of ACL	Surgical destabilization of the knee	2–3 weeks	Severe OA and subchondral bone destruction, though less rapidly than MIA model.	Technically difficult and time consuming.	[113,114,115]
DMM	Mouse	Transection of medial menisco-tibial ligament	Surgical destabilization of the knee	4–8 weeks	Modest OA, less rapidly than ACLT and MIA model. DMM model amenable to genetic manipulation.	Less difficult than ACLT	[107,112,116,117]

Abbreviation: Monosodium iodoacetate, MIA; Transection of the anterior cruciate ligament, ACLT; Destabilization of medial meniscus, DMM.

**Table 2 ijms-23-04642-t002:** Summary of new pharmacological therapies for OA pain in clinical trials.

Type of Drug	Drug Name	Route of Administration	ClinicalTrials.gov Identifier	Clinical Phase and Status	Affected OA Joint	Primary Measures or Results	References
CGRP antibody	Galcanezumab	S.C.	NCT02192190	Phase 2, completed	knee	No improvement in WOMAC pain	[60]
IL-1 receptor antagonist	Anakinra	I.A.	NCT00110916	Phase 2, completed	Knee	No improvements on OA symptoms	[72]
TNFα antibody	Adalimumab	S.C.	NCT00686439	Phase 2, completed	knee	Improvement in WOMAC pain.	[73]
NGF inhibitor	Tanezumab	I.V.	NCT00394563	Phase 2, completed	Knee	Reduction in joint pain and improvement in function	[120]
NGF inhibitor	Fasinumab	S.C.	NCT02709486, NCT02528188, NCT02697773	Phase 3, completed	Knee, hip	Improvement in pain within the first week, and pain and function were improved throughout 24 weeks	[121,122,123]
		S.C.	NCT02447276	Phase 3, completed	Knee, hip	Improvements in OA pain and function	[124]
		S.C.	NCT02683239, NCT03161093, NCT03304379	Phase 3, completed	Knee, hip	WOMAC pain subscale score	
	Tanezumab	I.V.	NCT00863304, NCT00830063, NCT00744471	Phase 3, completed	Knee, hip	Improvement of pain, physical function, and patient global assessment of OA	[125,126]
TrkA inhibitor	ASP7962	P.O.	NCT02611466	Phase 2, completed	Knee	No improvement in WOMAC pain	[127]
	GZ389988A	I.A.	NCT02845271	Phase 2, completed	Knee	Improvement in WOMAC pain	[128]
TNFα antibody	Adalimumab	S.C.	ACTRN12612000791831	Phase 2, completed	Hand	No improvements on symptoms or bone marrow lesions	[129]
		S.C.	NCT00597623	Phase 3, completed	Hand	No improvement in WOMAC pain	[130]
CCR2 antagonist	CNTX-6970	P.O.	NCT05025787	Phase 2, recruiting	Knee	WOMAC pain	
NGF/TNF-α bispecific antibody	MEDI7352	P.O.	NCT04675034	Phase 2b, recruiting	Knee	NRS	
IL-1α/β antibody	Lutikizumab	S.C.	NCT02384538	Phase 2, completed	Hand	No improvement in pain score	[131]
IL-1R1 antibody	AMG 108	I.V.	NCT00110942	Phase 2, completed	Knee	Minimal clinical benefit	[132]
IL-1α/β antibody	Lutikizumab	S.C.	NCT02087904	Phase 2, completed	Knee	No improvement in WOMAC pain	[133]
NLRP3 inhibitor	DVF890	P.O.	NCT04886258	Phase 2a, recruiting	Knee	KOOS pain sub-scale	
Wnt inhibitor	Lorecivivint (SM04690)	I.A.	NCT04385303 (NCT03928184 *)	Phase 3, active, not recruiting	Knee	Improvement in NRS Pain	[134]
TRPV1 inhibitor	CNTX-4975	I.A.	NCT03660943, NCT03661996	Phase 3, completed	Knee	WOMAC	
	NE06860	P.O.	NCT02712957	Phase 2, completed	Knee	NRS	
TRPA1 antagonist	LY3526318	P.O.	NCT05080660	Phase 2, recruiting	Knee	NRS and WOMAC pain	
SCN9A ^#^ antisense drug	OLP1002	S.C.	NCT05216341	Phase 2, recruiting	Knee, hip	WOMAC and VAS	
Central analgesic	Cannabinoid	P.O.	NCT04992962	Phase 2, recruiting	Knee	NRS and KOOS	
Tubulin inhibitor	Colchicine	P.O.	NCT03913442	Phase 4, recruiting	Knee	VAS pain scores	
Aryl hydrocarbon receptor antagonist	Resvertrol	P.O.	NCT02905799	Phase 3, recruiting	Knee	NRS	
Peroxynitrite decomposer	ACP044	P.O.	NCT05008835	Phase 2, recruiting	Knee	NRS	

* Terminated due to business reasons by Sponsor; ^#^ SCN9A gene encoding Nav1.7sodium ion channel; Abbreviation: Subcutaneous injection, S.C.; Intravenous injection, I.V.; Oral administration, P.O.; Intra-articular injection, I.A.; Numeric rating scale, NRS; Western Ontario and McMaster Universities Osteoarthritis Index, WOMAC; Visual analogue scale, VAS; Knee injury and osteoarthritis outcome score, KOOS.

## Data Availability

Not applicable.

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
