# Peer review of "Osteoarthritis Pain"

_ijms, 2022, doi:10.3390/ijms23094642_

Round 1

Reviewer 1 Report

“Osteoarthritis Pain” is an interesting review. However, I have several comments for the authors.

In general there are several sentences without references The authors should check and add appropriate references througout the manuscript.

The introduction on OA should be improved. OA is a whole joint disease involving all joint tissues: menisci, synovial membrane, infrapatellar fat pad and subchondral bone.

Lines 35-37: references should be provided.

Line 43-45: this part needs to be rewritten.

lines 45-54: References should be added.

Figure 1 should be not placed where the aim of the review is explained. Figure caption should be improved explaining the figure.

After the introduction, a brief paragraph about the methods should be added. Literature search database used, keywords used, criteria of exclusion/inclusion of the studies should be added.

Lines 65-78: references must be added for each concept.

Lines 79-80: again references must be added. For example, the possible contribution of the infrapatellar fat pad and synovial membrane on OA pain was discussed in a recent review.

Line 83: “sub-patellar fat” is unclear. Did the authors mean infrapatellar fat pad?

Lines 87-92: here, the authors claimed that the nature of pain includes three types: perceptive pain, inflammatory pain, and neuropathic pain and that OA pain is likely to be a mixture of the three. References must be provided.

Lines 89-94, Lines 113-116: again references must be provided.

Section 3: Since the authors focused on NGF, CGRP, CCL2/CCR2 and TNF-α, a brief introduction about these protiens could be useful at the beginning of each section.

Line 136: mouse model should be specified.

Lines 147-148: What is the potential protective pain?

Section 3.4: the authors should include IL-1beta in the subtitle.

Lines 209-216: references should be added. The inflammation also affects infrapatellar fat pad.

Lines 216-218: anti-cytokine therapies in clinical trials should be expanded.

Lines 225-226: “It is speculated that TNF-α and IL-1β maybe contribute differently in early stage and in late stage OA.” This sencente is unclear.

Lines 239-245, lines 273-277, lines 310-319: references are absent.

Section 3: tables summarizing the results should be added.

Section 4: the title of this section needs to be improved. A table summarizing MIA-induced OA model and DMM surgery should be added. What about other OA models? For example, what about anterior cruciate ligament transection (ACLT) model?

Table 1 should be contextualized. It is not acceptable to cite a table in the title of the section.

Data reported in table 1 should be follow the same order of data presented in the text.

Lines 356: here the authors mention erosive hand OA. However, this type of OA was never introduced. Since this journal is not a rheumatologic/orthopedic specific journal, a very brief introduction (even if the review is not focused on this particular type of OA) should be added (this recent review could be useful: DOI:10.1038/s41584-021-00747-3).

IL-1beta antibodies should be added.

Contracted forms such as “doesn’t” should not be used.

All abbreviations should be defined at first mention.

Reviewer 2 Report

This review manuscript about OA pain is overall well written and scientific concepts are well explained. References are cited in a proper manner and are reasonably recent to still be of value in the field.

Therefore, I think that no particular changes in the manuscript are required.

Just a couple of things needs to be specified: 

Pag 2 line 71: I think a reference is needed (especially to introduce a concept)

pag 3 line 82: the sentence has to be rephrased (it does not make sense as an english sentence)

Round 2

Reviewer 1 Report

The manuscript improved after the revision.

No additional comments.

This manuscript is a resubmission of an earlier submission. The following is a list of the peer review reports and author responses from that submission.